# Tissue-Specific Regulation of *CFTR* Gene Expression

**DOI:** 10.3390/ijms241310678

**Published:** 2023-06-26

**Authors:** Clara Blotas, Claude Férec, Stéphanie Moisan

**Affiliations:** 1Univ Brest, Inserm, EFS, UMR 1078, GGB, F-29200 Brest, France; clara.blotas@univ-brest.fr (C.B.); claude.ferec@univ-brest.fr (C.F.); 2Laboratoire de Génétique Moléculaire et d’Histocompatibilité, CHU Brest, F-29200 Brest, France

**Keywords:** cystic fibrosis, *CFTR*-associated disease, *cis*-regulatory elements, microRNA, modifier genes, tissue-specificity, chromatin organization

## Abstract

More than 2000 variations are described within the *CFTR* (*Cystic Fibrosis Transmembrane Regulator*) gene and related to large clinical issues from cystic fibrosis to mono-organ diseases. Although these *CFTR*-associated diseases have been well documented, a large phenotype spectrum is observed and correlations between phenotypes and genotypes are still not well established. To address this issue, we present several regulatory elements that can modulate *CFTR* gene expression in a tissue-specific manner. Among them, *cis*-regulatory elements act through chromatin loopings and take part in three-dimensional structured organization. With tissue-specific transcription factors, they form chromatin modules and can regulate gene expression. Alterations of specific regulations can impact and modulate disease expressions. Understanding all those mechanisms highlights the need to expand research outside the gene to enhance our knowledge.

## 1. Introduction

In 1595, the first historical record of Cystic Fibrosis (CF) was made. CF was described as a nutritional disturbance with diarrhea, steatorrhea, growth failure, salty sweat, bronchopneumonia and pulmonary infection [1,2]. Based on family studies, Andersen and May defined CF as a genetic disorder with autosomal recessive transmission [3,4]. In 1989, molecular biology development led to the discovery of the *CFTR* (*Cystic Fibrosis Transmembrane Regulator*) gene and also rapidly to the first report of a variation, F508del, which is now identified in 70% of people with CF (pwCF) [5,6,7,8]. The CFTR protein structure is described as an ion channel capable of modulating NaCl efflux. Genotyping in pwCF led to the identification of 2114 variations in the *CFTR* gene (genet.sickkids.on.ca, accessed on 21 May 2023). However, it is still very difficult to establish a link between the phenotype of individuals with *CFTR*-associated diseases and their genotype. Much remains to be done in the field to better understand the implication of the *CFTR* gene.

This review aims to decipher the implication of regulatory elements in the *CFTR* gene expression and hence the modulation of disease expression. *Cis-* and *trans*-regulations are two mechanisms discussed here. 

## 2. Clinical Issues

Cystic fibrosis is a monogenic disorder with autosomal recessive transmission. It is caused by alterations of the *CFTR* gene, which spans over 189 kilobases with 27 exons and encodes a protein of 1480 amino acids. The CFTR protein belongs to the ATP-binding cassette (ABC) transporter family, which acts as an AMPc regulated ion channel. Many variants are found within the *CFTR* gene and are classified in seven classes depending on their effect on the protein (no protein, less protein, impaired gating, etc.) [9]. The most common variation found in 70% of pwCF is the F508del.

Defects of the CFTR protein lead to dysfunctions of the channel in epithelial cells. Ion transports are affected, resulting in water absorption into mucus. Obstruction of organs expressing CFTR is the consequence of mucus thickening.

pwCF are mainly affected by respiratory tract damage. In addition to the obstructive aspect, mucus thickening promotes bacterial development leading to important pulmonary infection, which is the major cause of mortality. Cystic fibrosis is a multi-organ disorder. pwCF are also affected by exocrine pancreatic damage, intestinal obstruction, infertility and liver and bone diseases [10].

However, the clinical spectrum is complex as variations in the *CFTR* gene also cause several mono-organ disorders, such as pancreatitis or congenital bilateral absence of the vas deferens (CBAVD) [10]. CBAVD represents 3% of male infertility and is the most frequent *CFTR*-related disorder (*CFTR*-RD) [11]. Two genes have been implicated, the *CFTR* gene in 80% of cases and *ADGRG2* [12]. ADGRG2, a G protein-coupled receptor, regulates fluid reabsorption in efferent ducts through the activation of *CFTR* [13]. Variants in *CFTR* lead to a defect in HCO_3_^−^ secretion, which has been shown to be critical for fertilizing the capacity of sperm [14]. Pancreatitis is also caused by a defect of HCO_3_^−^ secretion, essential to solubilize mucins and avoid the plugging of pancreatic duct [15].

The *CFTR* gene is expressed in many organs with different levels of expression, and molecular consequences of a defect are variable across tissues. We, therefore, have a complex picture with a variable genotype–phenotype relationship.

## 3. Tissue Expression

Clinical issues of *CFTR*-associated diseases are directly correlated with the gene expression pattern. Indeed, a very clear tissue-specificity expression is observed but is also a temporal aspect. In-situ hybridizations show that *CFTR* is expressed in particular in epithelial cells with a very great level in the pancreatic duct and nasal polyps and to a lesser extent in lungs, gut, sweat glands, placenta, liver and male genital ducts as shown in Figure 1 [6]. In the male reproductive system, the *CFTR* gene is in majority expressed in vas deferens [16]. After birth the *CFTR* mRNA level is low, in particular in lungs, which is surprising in view of lung disease lethality. In fact, unique rare epithelial cells within the lung express the highest level of *CFTR*, the ionocytes [17,18]. More recently, secretory cells have been identified as the most important cells that express *CFTR* in lungs [19].

## 4. Characteristics of the *CFTR* Locus

### 4.1. Promoter as a Housekeeping Gene

For the majority of genes, the regulation of its transcription relies on the binding of specific transcription factors, activators or repressors, to the promoter. Surprisingly, while the *CFTR* gene expression is strictly regulated both spatially and temporally, its promoter has many features of a housekeeping gene. The *CFTR* promoter has no TATA box and possesses high GC content and multiple transcription start sites [20,21]. Several binding sites for transcription factors are observed as activator protein 1 (AP-1), Sp1, cAMP response element binding protein (CREB), an inverted CCAAT element (Y-box) and a CArG-like motif [22]. Also, an iκβ-like motif allows NF-κB binding next to inflammatory context and a HIF Responsive Element (HRE) motif in case of hypoxia (Figure 2) [23,24].

However, these elements alone cannot explain the precise tissue-specificity of the *CFTR* gene.

### 4.2. Cis-Regulation of the CFTR Gene

While elements within the *CFTR* promoter cannot explain its regulation, some research teams decided to study long range elements to provide a clearer picture. Chromatin in the nucleus is highly organized in different compartments. Long-distance elements can interact with each other through chromatin looping, notably within topologically associated domains (TAD). A huge effort to understand the non-coding DNA has been made through the ENCODE project, leading to a better characterization of *cis*-regulatory elements (CREs) [25]. CREs are regularly identified within DNase I hypersensitive sites (DHS); overlap biochemical marks (H3K27ac, activator or H3K27me3, repressor); and bind functional elements. They can interact with promoter or other CREs [26]. If functional characterizations do not provide all features, an element is just described as candidate CRE (cCRE). Activities are context-dependent as *cis*-regulation is tissue-specific.

The *cis*-regulation of the *CFTR* locus began to be explored since the 90s, and several elements are well described. Some elements are common to all cell types expressing *CFTR* and are considered more as structural elements, and some are more related to tissue-specific expression. A *CFTR* TAD was described as spanning 317 kb (chr7:117,039,878–117,356,812, hg19) and including the neighboring genes *ASZ1* and *CTTNBP*2. Boundaries are delineated by two CREs identified as barrier insulators illustrated in Figure 3, in 5′ at −80.1 kb from the transcription start site (TSS) and in 3′ at +48.9 kb from the last codon of *CFTR* [27,28]. Another type of insulator exists: the enhancer-blocking insulator. Its role is to block interactions between promoter and CREs. Three enhancer-blocking insulators are described in the TAD; at −20.9 kb and +6.8 kb the activity is related to the recruitment of the CTCF protein and at +15.6 kb by the recruitment of C/EBP, CREB, AP-1, ARP-1 and HNF-4 [29,30].

#### 4.2.1. Airway

Several DHS have been identified in different pulmonary cells across the *CFTR* locus and have been shown to interact with the promoter. DHS at −35 kb from the TSS and DHS at −44 kb are the most evident regions that interact with the promoter in airway epithelial cells [28,31]. H3K27ac marks correlating with transcriptionally active chromatin overlap with these two regions. In the presence of both elements, the *CFTR* promoter highly increases its activity (×30), demonstrating a cooperative effect [32]. Furthermore, RNA polymerase II binds these elements, which may imply a production of enhancer RNA (eRNA) [33]. eRNA are unstable transcripts emerging from active enhancer and acting as transcriptional regulators [34]. Identifying transcription factors that bind CREs helps to explain the three-dimensional organization and the formation of DNA looping. At −35 kb, a complex of transcription factors has been identified by ChIP-seq (Chromatin Immunoprecipitation Sequencing), including BAF155 (subunit of nucleosome remodeling complex SWI/SNF), IRF1 (interferon regulatory factor 1), NF-Y (Nuclear transcription factor Y), EHF (ETS Homologous Factor) and KLF5 (Kruppel-like factor 5) [35,36,37]. A nucleosome depletion of this region has also been shown, confirming its accessibility [35]. At −44 kb, recruitment of BACH1 (BTB Domain and CNC Homolog 1), BRD8 (Bromodomain Containing 8) and CTCF have been described [38,39]. Those two CREs are essential, a dramatic loss of *CFTR* expression is observed after their depletion as well as a loss of three-dimensional organization of the locus [33].

In addition to these two CREs, CREs at −3.4 kb from the TSS and in the intron 26 (chr7:117,305,325–117,307,149, intron 23 legacy name) show an increase of 3-fold of promotor activity in a cell-type specific manner [31]. A cooperative effect is observed when CREs −35 kb, −3.4 kb and intron 26 are associated. A weak enhancer effect is observed in the presence of another region at +36.6 kb from the last codon, but it increases in association with CRE −44 kb [40]. Some other regions are implicated in the maintenance of the locus architecture, for example a CTCF site at −20.9 kb [27,29]. An element in intron 22 (chr7:117,280,467–117,283,006) interacts with the promoter but without effect on the promoter activity [40].

Hence, many CREs appear to be involved in the correct expression of the *CFTR* gene in airway epithelial cells. For a correct level of expression, H3K4 mono-methylation by methyltransferase SEDT7 (SET Domain Containing 7) is required at CRE −35 kb for the binding of NF-Y. Depletion of SETD7 shows a loss of NF-Y binding and prevents occupation of SIN3a (SIN3 Transcription Regulator Family Member A) causing an enrichment of p300 (p300 histones acetyltransferase) at multiple sites including −44 kb and thus an increase of *CFTR* expression [36]. These observations suggest the presence of control mechanisms when a CRE is defective. When the KLF5 repressor is depleted, *CFTR* expression increases by six-fold as well as the activity of the ion channel [37]. Conversely, when IRF1 is depleted, the expression of the *CFTR* gene decreases [36].

Briefly, CREs at −44 kb and −35 kb are two strong enhancers which interact with the promoter and other CREs at −3.4 kb, in intron 26 and in 5′ of the gene at +36.6 kb to accent the effect. A complex of transcription factors binds these elements to ensure the correct level of gene expression. To maintain this organization, structural elements are used such as the −80.1 kb and +48.9 kb TAD boundaries and the −20.9 kb CTCF site (Figure 4).

#### 4.2.2. Intestine

In intestinal cells, the same type of specific *cis*-regulations are observed. DHS are observed along the locus, and cell type-specific elements have been identified. By chromatin conformation capture studies, DHS in the introns 1, 11 and 12 (chr7:117,129,649–117,130,749; chr7:117,212,364–117,213,789; chr7:117,227,802–117,229,475, respectively), overlapping H3K27ac marks, have been shown to interact with the *CFTR* promoter [41,42,43]. DHS in 5′ of the gene, −80.1 kb and −20.9 kb as well as downstream in 3′, +48.9 kb, also interact with the promoter [27,28]. DHS in introns 1 and 12 are the most obvious CREs implicated in the *CFTR cis*-regulation in the small intestine. CRE in intron 12 is a strong enhancer, and in combination with CRE in intron 1, the effect is doubled (×33) [41,44]. Assuming that cooperation is frequently used, other combinations have been made in Caco2 cells, the most frequent intestinal cell line used. When CRE in intron 12 and the enhancer-blocking insulator at +15.6 kb are combined, a more impressive effect is observed [41]. DHS in intron 24 and 26 (chr7:117,299,158–117,301,081; chr7:117,305,325–117,307,149) are present but have weak enhancer activities on the promoter. However, if they are combined with intron 12, the enhancer activity increases by 57-fold and 34-fold, respectively [44]. When the three regions are combined, the same effect as CREs 12–24 is observed. Many transcription factors bind the different CREs notably because sequences in intron 1 and 12 are nucleosome-free regions [35]. Intron 12 is bound by RNA polymerase II [27]. Transcription factors HNF1α (hepatocyte nuclear factor 1 homeobox A), CDX2 (Caudal Type Homeobox 2), HNF4 (Hepatocyte Nuclear Factor 4) and FOXA2 (Forkhead Box A2) form a complex and bind CREs 1, 11, 12, 24 and 26 [44]. This high local concentration of transcription factors can illustrate a chromatin module [45]. HNF1α acts in this case as a master regulator and certainly stabilizes the binding of transcription factors. Besides, its expression correlates with the *CFTR* gene expression [46]. Hence, a variation in the HNF1α binding site leads to a loss of binding of the protein complex; notably it is observed that, in absence of HNF1α, a loss of acetylation occurs, leading to a decrease of *CFTR* expression [42,47]. In addition, TCF4 (Transcription Factor 4) binds CREs in intron 1 and 24, and its deletion also leads to a decrease of *CFTR* expression [47]. A loss of FOXA2 perturbs chromatin interactions, indeed its role is to maintain open chromatin [48].

If CRE in intron 1 is depleted, the *CFTR* gene expression decreases by 60% in vivo [49]. When the CREs 1 and 12 are depleted, the *CFTR* gene expression is almost undetectable [48]. After CRISPR/Cas9 deletions of CRE in intron 1 and 12, great changes in the three-dimensional organization are observed [48]. A gain of interactions between the *CFTR* promoter and many regions of the locus is observed, in particular with TAD boundaries, CTCF sites at −20.9 kb and +6.8 kb and intronic CREs as intron 4, 12 and 26 next to the deletion of intron 1. In the case of a loss of intron 12, interactions decrease between promoter and introns 1 and 11. A double deletion leads to the loss of all interactions between promoter and regions around both CREs and significantly disrupts the higher order chromatin organization, suggesting the very important role of these elements [48].

CRE in intron 1 is a weak enhancer but has a more important role in maintaining the locus architecture, while CRE in intron 12 is a strong enhancer and acts directly on *CFTR* expression (Figure 5).

#### 4.2.3. Epididymis

In epididymis cells several DHS have been described in 5′ at −20.9 kb, in introns 1, 23 and 26 and in the 3′ region at +15.6 kb and +6.8 kb [29,41,50,51]. Also, a more distal region at the 3′ end of *WNT2* gene is described. Active chromatin marks as H3K4me1 and H3K27ac are particularly present within DHS23 [52]. Nucleosome depletion occurs also at this site and c-Fos, c-Jun, JunD and C/EBP are predicted to bind the region [52]. Moreover, this region slightly interacts with the promoter. In addition, DHS 2 and 4 as well as 5′ regions and DHS + 48.9 kb interact with the promoter [27,53]. The +6.8 kb region binds CTCF and acts as an enhancer-blocking insulator as well as the +15.6 kb but without recruitment of CTCF protein [29]. Unlike the other cell types, the strongest interactions with the promoter are in 5′ regions at −80.1 kb and −20.9 kb. Reporter gene assay has not been performed; the majority of the data being produced with primary cells and no relevant cell lines are available.

HNF1α seems to have an important role in the regulation of genes expressed in the epididymis including *CFTR*. Indeed, a depletion by siRNA induces a decrease of genes implicated in ion transports, such as the solute carrier family. Perturbation of the regulation via HNF1α disturbs the intracellular pH, and finally, the luminal environment is modified [54]. ChIP-seq data indicate the binding of HNF1α at −44 kb and +15.6 kb [54].

Regions upstream and downstream the *CFTR* gene seem to have an important role as well as the region in intron 23 in regulating expression in epididymis cells (Figure 6).

#### 4.2.4. Pancreas

The *cis*-regulation of the *CFTR* gene in the pancreas is less well-known, certainly because of the lack of relevant cell models. In fact, the duct pancreatic cells highly express the *CFTR* gene while cell lines available express very low levels of it. Hence, only candidate CREs have been highlighted. DNase-seq data indicate DHS in 5′ end of the gene, at −80.1 kb, −44 kb, −35 kb and in the 3′ at +6.8 kb and +15.6 kb [55]. The most important regions seem to be intronic. DHS have been observed at introns 2, 18, 19, 21 and 23. By EMSA (Electrophoretic Mobility Shift Assay) bindings of HNF1α, CDX2 and PBX1 (PBX Homeobox 1) have been shown on DHS 18 and 19, but only PBX1 has been confirmed in vivo [56]. Nevertheless, HNF1α depletion leads to *CFTR* mRNA decrease. Another transcription factor positively impacts the *CFTR* expression, BAF155, involved in nucleosome remodeling [57]. DHS 21 activates *CFTR* transcription next to mitomycine C treatment, an activator of *CFTR* [58].

To go farther, Smith et al. have achieved 5C (Chromosome Conformation Capture Carbon Copy) experiments on Capan-1 (pancreatic cell line) and showed a unique important interaction between the *CFTR* promoter and the region in intron 11. The interaction profile is rather low and looks like interaction profiles of cells that do not express *CFTR*. However, TAD boundaries are conserved [28]. This confirms that TADs are conserved between cell types but that intra-TAD structures vary according to the context.

Only candidate CREs have been described in pancreatic cells with assumptions allowing to implement a three-dimensional *cis*-regulatory model (Figure 7). Studies have to be carried out to obtain an accurate model.

### 4.3. Impact of CFTR Cis-Regulatory Variants?

It is now well-known that, in presence of alterations within CREs, diseases can occur; this is defined by the term enhanceropathies [59].

Next to the analysis of 210 kb across the *CFTR* locus on F508del homozygous patients, several variants have been newly identified, and some of them are found within CREs as −80.1 kb, −44 kb, −35 kb, −20.9 kb, intron 12 and +48.9 kb [60]. Association studies on lung function or sweat chloride concentrations have highlighted that common or rare variants could modulate the CF phenotype (positively or negatively). The most striking effect for both conditions is caused by variants within −80.1 kb, which corresponds to a *CFTR* TAD boundary. Important structural modifications can induce a change of expression level and partially recover a sufficient amount of CFTR at the membrane. Not surprisingly, variants present in CREs, interacting with the promoter only in epididymis cells, have no significant effect on lung function or sweat chloride level.

In 2019, Kerschner et al. studied 80 patients with 1 or 2 unknown *CFTR* variants and performed a resequencing of a region of 463 kb [61]. Variants (total: 1737) have been identified, and one-half of the studied alleles contains unknown pathogenic variants. Variants (51) are localized among 17 CREs (37 substitutions, 11 indels). Some of them are present in addition to two causal variants. In regions −44 kb, −35 kb, introns 1, 12 and 26, variants have been found within transcription factor binding sites. Four variants localized in CRE of intron 11 induced a decrease from 37% to 63% of promoter activity by luciferase assays. More functional tests are required to validate the impact of identifying variants, but this study demonstrates the importance of variants in non-coding DNA.

Several variants identified within CREs have led to modification in chromatin organization or in promoter activity in a tissue specific manner, showing the importance of investigating these regions.

### 4.4. microRNA and lncRNA

MicroRNA or miRNA are small non-coding RNA capable of modulating gene expression through the binding of 3′UTR of the targeted gene. As well as CREs, miRNA are expressed in a tissue-specific manner. Several miRNA have been highlighted to impact *CFTR* expression directly or indirectly by binding genes regulating *CFTR* [62].

miR-145, miR-331-3p and miR-494 are the first described miRNA to regulate *CFTR* in pulmonary cells [63]. More precise study of temporal regulation shows that miR-145 and miR-101, actually, have no effect in fetal lung cells but regulate negatively the level of *CFTR* mRNA in adult lung cells [64]. Blocking both miRNAs’ binding sites allows the stabilization of *CFTR* mRNA, hence increased channel activity of F508del homozygous patients [64]. miR-145 and miR-494 are also involved in *CFTR* regulation in intestinal cells by repressing by 40% *CFTR* mRNA [63]. In pancreatic cells, miR-1246, miR-1290 and miR-1827 repress *CFTR* levels [63].

The miR-888 cluster is exclusively expressed in the reproductive tract, in particular in epididymis, but without link with *CFTR* [65]. Other miRNA have also been described in different parts of the epididymis as miR-573, miR-155, miR-1204 and miR-770 [66].

In contrast to microRNA, long non-coding RNA (lncRNA) are made up of more than 200 nucleotides and constitute a group of non-coding DNA which is not yet well described [67]. A study comparing bronchial brushings from pwCF and non-CF identifies 1063 lncRNAs differently expressed [68]. Next to Pseudomonas aeruginosa (PsA) infection, a modification in several lncRNA as MEG9 (Maternally Expressed 9), a positive co-regulator of inflammatory pathways in CF lung, is also observed [69]. Functional tests on BGas lncRNA, described by Saayman et al., show that its transcription is initiate from intron 11 of the *CFTR* gene and regulates negatively its expression by acting with other protein and modifying local chromatin conformation [70].

Both microRNA and lncRNA belong to the family of non-coding DNA, previously named “Junk DNA”. Here, we report few examples that highlight the importance of those elements that are capable of regulating the *CFTR* gene with varying degrees of tissue specificity.

### 4.5. Modifier Genes

Other genetic factors can impact the *CFTR* gene. In fact, each individual is characterized by a specific polymorphism pattern. Polymorphism is a common variant in DNA sequence with an allele frequency of at least 1% in the general population. The presence of a single nucleotide polymorphism (SNP) alone is not pathogenic, but in a specific gene it can modulate the severity of a disease caused by another gene; they are referred as modifier genes. In CF and *CFTR*-RD, several genes have been implicated to modify lung function, the bacterial infection and inflammation and the severity of the intestinal obstruction.

#### 4.5.1. Pulmonary

Huge genome wide association study (GWAS) identifies several locus/modifier genes that could impact lung function [71,72]. For example, natural host defense barrier can be altered in case of variations within MUC4/MUC20. These are glycoproteins, mucins, expressed on the surface of epithelial cells from gastrointestinal and respiratory tracts that prevent mucus accumulation in periciliary layer. Hence, they represent a host defense barrier and are essential to maintaining equilibrium [73]. SLC9A3 is an ion transporter acting on pH regulation and is involved in neonatal intestinal obstruction. Early infection by PsA is related with *SLC9A3* variants [74,75]. Defect on pH regulation leads to depletion of airway liquid surface representing a supportive environment to bacterial growth. On the same locus, *EXOC3* is part of post-Golgi trafficking. Another locus contains the modifier gene *HLA* class II involved in immune response, which is associated with asthma and altered lung function. *AGTR2* is associated with lung fibrosis and *SLC6A14* with lung severity and appearance of PsA [76]. *EHF* has an important role in lung epithelial function by regulating inflammation and pathways in response to injury [77]. CE72/TPPP plays a role in microtubule organization, which is disturbed in CF [71]. Cytokines as TGFβ1, IL-8, Il1β, TNFα are related to inflammation in CF [78].

#### 4.5.2. Intestines

Meconium ileus, which presents in ~15% of pwCF, is also highly related to modifier genes, such as the solute carrier family as *SLC26A9*, *SLC9A3*, *SLC4A4* and *SLC6A14*, *ATP12A* and *PRSS1* [78,79,80]. Several proteins from the solute carrier family have been shown to interact with CFTR and to impact its activity. We can suppose that the presence of SNPs within a gene mentioned above can modify this interaction. In the case of *ATP12A,* a proton exchanger, minor changes in the ion transport could affect viscosity [80].

#### 4.5.3. Epididymis

In the case of infertility due to absence of vas deferens, fewer studies have been performed. *TGFβ1*, *EDNRA* and *SLC9A3* were only suggested to impact the correct development of the reproductive male tracts [81,82,83].

#### 4.5.4. Pancreas

Pancreatic insufficiency in CF is also probably related to modifier genes, and some of them have been predicted but have not been confirmed by functional studies. *CTNNB1*, *IRF5*, *EPHX1*, *PRSS1*, *CASR*, *CTRC* and *KRT8* are some of them [84,85,86,87].

Modifier genes seem to have an important role in the different tissues to modify the phenotype in addition to the CFTR defect. Huge efforts need to be taken to accurately understand the functional linkage with phenotype.

### 4.6. Complex Alleles

In 2018, the Cystic Fibrosis Foundation Patient Registry reported that 3.1% of pwCF carry more than two *CFTR* variants. Complex allele is defined by the presence of an additional variation on the same parental allele. The presence of one more variant can modulate the CFTR function, and several combinations are reported in literature. For example, a severe effect on CFTR function is observed in the presence of complex allele c.[1397C  >  A;3209G  >  A] [88]. In the case of c.3469-1304C > G, all the patients present early symptoms with PsA colonization and pancreatic insufficiency [89]. Conversely, patients with c.3874-4522A > G present fewer symptoms with moderate effect, corresponding to *CFTR*-RD characteristics [89]. To gain insight into the understanding of complex alleles a recent protocol of Targeted Locus Amplification and Haplotyping applied to the *CFTR* gene has been published [90].

It is important to gain insight about complex alleles. Complex alleles present within CREs can modify a phenotype due to tissue-specific activation of them. If we limit ourselves to identifying just two variations, we may miss the element explaining different phenotypes. Complex alleles must be taken into serious consideration, in particular for the implementation of treatment.

## 5. Conclusions

Care for pwCF has largely improved these past years through the development of modulators (elexacaftor/tezacaftor/ivacaftor) treatment [91,92]. Nonetheless, some patients are still not eligible or not responding. Moreover, some moderate *CFTR* disease cases remain misunderstood. These different findings highlight the necessity to understand the precise mechanism underlying the different patterns of disease’s expression and the relationship between genotype and phenotype.

Multiple elements are implicated in the modulation of diseases in a tissue-specific manner but also in the response of treatment. Actually, numerous pathways can impact the *CFTR* expression through indirect cellular regulations as inflammation, hypoxia, oxidative stress and endoplasmic reticulum stress or through transcriptional regulation [23,24,93]. Here, we decided to list some elements that have been demonstrated to impact *CFTR* transcriptional expression and detail different regulations in few cell types. We have not mentioned *CFTR* expression in non-epithelial tissues such as neutrophils, macrophages, brain, bone due to the lack of available data. We present modifier genes as an important mechanism that can modify specific clinical manifestations, in particular according to the tissue expression of the modifier gene. We refer to the presence of complex alleles that are probably underestimated but which must definitely affect responses to treatments [60]. Some complex alleles have been identified as non-responder [94]. Also, we highlight the importance of knowing their presence and thus explaining phenotype if, for example, active CREs are affected by them. This leads to the last element described in the review, CREs. They act at distance from the gene to regulate its expression, and variants can lead to dysfunction. Functional tests have to be performed to confirm effects of the *cis*-regulatory variants found in patients. This also highlights the need to perform a resequencing on larger regions in unresolved cases. Also, we stress that three-dimensional organization is very important and highly tissue-related depending on individual transcription factors recruitment. Alterations of *cis*-regulation mechanism lead to various clinical manifestations.

This review states diverse gene regulation systems and outlines the importance to understand basal gene expression in the different tissues and to transfer this knowledge to the clinical management.

## Figures and Tables

**Figure 1 ijms-24-10678-f001:**
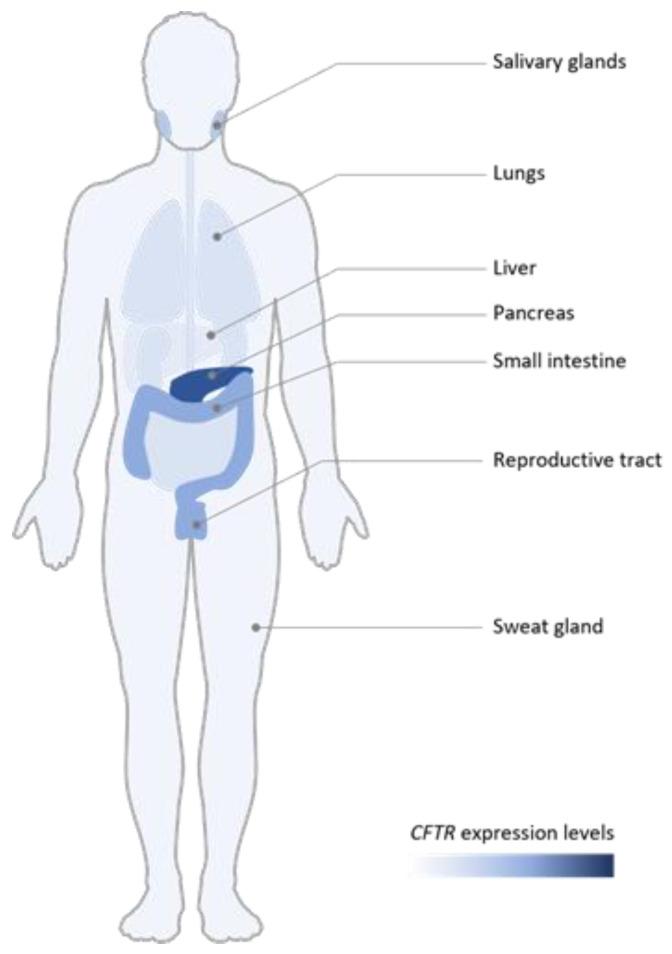
Differential expressions of the *CFTR* gene in a human. High *CFTR* expressions are detected in reproductive tract, pancreas, small intestine and salivary glands. Lower levels are observed in liver and lungs.

**Figure 2 ijms-24-10678-f002:**
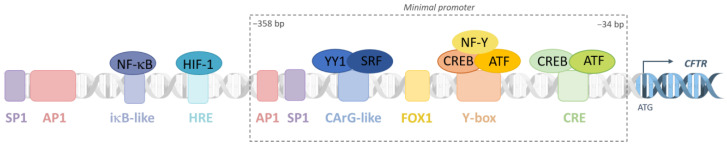
Proximal regulatory elements of the *CFTR* promoter. The *CFTR* translation start site is represented by the ATG codon, and upstream different motifs are present. The minimal promoter starts at −34 bp from the ATG codon to −358 bp. To allow correct expression of the *CFTR* gene, the promoter is composed of several regulatory elements, such as a Cyclic AMP Response Element motif, a Y-box motif (inverted CCAAT element), a CArG-like motif, binding motifs for Activator Protein-1 (AP-1), Specificity Protein 1 (SP1), an iκβ-like motif and a HIF Responsive Element (HRE) motif.

**Figure 3 ijms-24-10678-f003:**
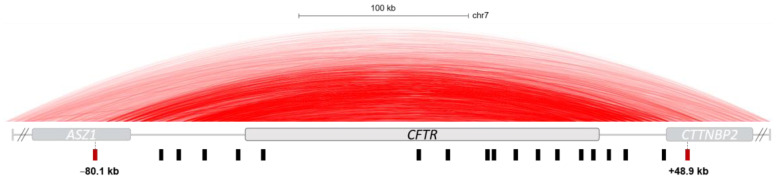
*Cis*-regulatory elements across the *CFTR* locus. Hi-C data from UCSC genome browser (IMR90 cells—hg19) show a *CFTR* TAD. This TAD is delineated by boundaries at −80.1 kb of TSS of the *CFTR* gene in *ASZ1* gene and +48.9 kb of the last codon of *CFTR* gene in *CTTNBP2* gene. Boundaries are represented by red squares and CREs by black squares.

**Figure 4 ijms-24-10678-f004:**
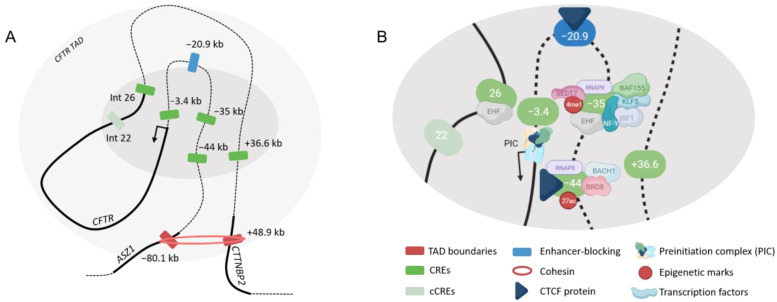
*Cis*-regulation of the *CFTR* gene in airway epithelial cells. Three-dimensional representation of the *CFTR* locus (**A**). Chromatin loopings allow CREs to interact with the *CFTR* promoter. Coding sequences are represented in thick lines. Chromatin module (**B**) with multiple transcription factors binding specific CREs. −44 kb and −35 kb are the most implicated CREs in airway epithelial cells.

**Figure 5 ijms-24-10678-f005:**
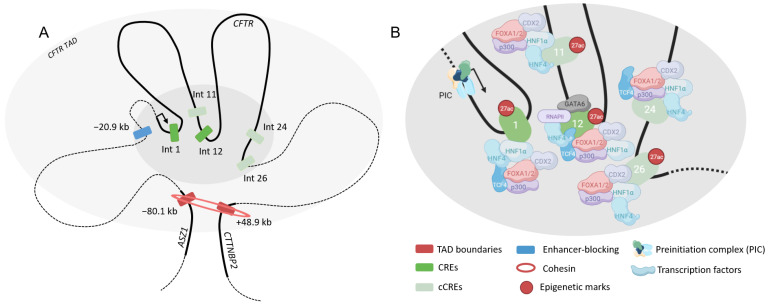
*Cis*-regulation of the *CFTR* gene in intestinal cells. Three-dimensional representation of the *CFTR* locus (**A**). Chromatin loopings allow CREs to interact with the *CFTR* promoter. Coding sequences are represented in thick lines. Chromatin module (**B**) with multiple transcription factors binding specific CREs. CREs in introns 1 and 12 are the most implicated CREs in intestinal epithelial cells.

**Figure 6 ijms-24-10678-f006:**
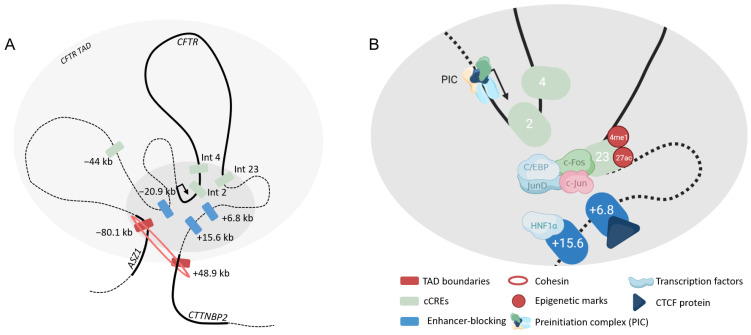
*Cis*-regulation of the *CFTR* gene in epididymis cells. Three-dimensional representation of the *CFTR* locus (**A**). Chromatin loopings allow CREs to interact with the *CFTR* promoter. Coding sequences are represented in thick lines. Chromatin module (**B**) with multiple transcription factors binding specific cCREs. Only predicted regions are highlighted.

**Figure 7 ijms-24-10678-f007:**
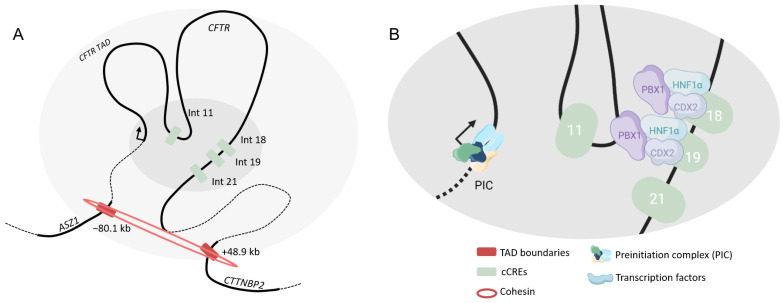
*Cis*-regulation of the *CFTR* gene in pancreatic cells. Three-dimensional representation of the *CFTR* locus (**A**). Chromatin loopings allow CREs to interact with the *CFTR* promoter. Coding sequences are represented in thick lines. Chromatin module (**B**) with multiple transcription factors binding specific cCREs. Only predicted regions are highlighted.

## Data Availability

Not applicable.

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
