# Peer review of "Tissue-Specific Regulation of *CFTR* Gene Expression"

_ijms, 2023, doi:10.3390/ijms241310678_

Round 1

Reviewer 1 Report

In this article, Blotas et al review many aspects on “CFTR tissue-specific regulations”.

The laboratory is very well known and leader in the field.

The review is well written, accurate and very complete. Figures illustrate the text appropriately. I would suggest some modifications to improve the manuscript.

1. Add a small conclusion at the end of the different sections

2. After the Tissue expression section, please add a paragraph on pathways that have been shown to regulate CFTR expression, such as inflammation, hypoxia…

2. CFTR promoter section: add NF-kB which links CFTR expression to inflammation and add a figure illustrating the elements identified within the CFTR promoter

2. microRNA section: add a paragraph on CFTR regulation by LncRNAs within this section

5. Conclusion: add in the conclusion that CFTR has also been detected in non-epithelial tissues, such as neutrophils, macrophages, brain, bone… and that tissue-specific expression in these tissues is not described

Minor

1. p2 line 10: liver and bone re also affected

P2, line 55: develop the known role of ADGRG2 in the occurrence of CBAVD.Is there a link with CFTR?

P4, line 124: explain more “enhancer RNA (eRNA)”

P4, line 152: does KLF5 depletion also affect CFTR expression or only channel activity? Pleases precise

P5, line 176: state that Caco2 cells are of intestinal origin and frequently used as a model

Author Response

We thank the Referee for his/her evaluation, and for appreciating the quality of our review. Our answers to their comments are found below.

Comments:

  1. Add a small conclusion at the end of the different sections

Response: We agree with the Referee for this note and the importance of concluding the different sections and modified the text to include the following sentences:

"Page 7, Lines 271-274

"Page 8, Lines 301-303

"Page 9, Lines 329-331

"Page 10, Lines 374-376

"Page 10, Lines 389-393

  1. After the Tissue expression section, please add a paragraph on pathways that have been shown to regulate CFTR expression, such as inflammation, hypoxia…

Response: We understand this comment and agree with the Referee that post-transcriptional regulations with different pathways such as inflammation, hypoxia… are also very important for CFTR regulation. In our opinion, as our review focuses on CFTR transcriptional regulation mechanisms, we decided to mention these other pathways within the discussion.

"Page 10, Lines 396-398

“Multiple elements are implicated in the modulation of diseases, in a tissue-specific manner, but also in the response of treatment. Actually, numerous pathways can impact the CFTR expression through indirect cellular regulations as inflammation, hypoxia, oxidative stress and endoplasmic reticulum stress or through transcriptional regulation. Here, we decided to list some elements which have been demonstrated to impact CFTR transcriptional expression and detail different regulations in few cell types.”

  1. CFTR promoter section: add NF-kB which links CFTR expression to inflammation and add a figure illustrating the elements identified within the CFTR promoter

Response: We agree with the Referee and apologize for not indicating it. We completed this CFTR promoter section and added a figure illustrating the elements identified within the CFTR promoter.

"Page 3, Lines 88-89

  1. microRNA section: add a paragraph on CFTR regulation by LncRNAs within this section

Response: We understand the suggestion and include explanation about lncRNA into the microRNA section.

"Page 9, Lines 320-328

  1. Conclusion: add in the conclusion that CFTR has also been detected in non-epithelial tissues, such as neutrophils, macrophages, brain, bone… and that tissue-specific expression in these tissues is not described

Response: We thanks the Reviewer for his comment and complete the conclusion.

Minor revisions :

  1. p2 line 10: liver and bone re also affected

Response: We concur and modified the text to also mention them.

"Page 2, Lines 50-51

  1. P2, line 55: develop the known role of ADGRG2 in the occurrence of CBAVD.Is there a link with CFTR?

Response: We thank the Reviewer for this comment and modified the text to clarify this statement with the following sentence and further reference.

“ADGRG2, a G protein-coupled receptor, regulates fluid reabsorption in efferent duct through the activation of CFTR”

  1. P4, line 124: explain more “enhancer RNA (eRNA)”

Response: The reviewer is correct that we could more explain these eRNA which are important elements. We added this sentence in the revised manuscript.

"Page 4, Lines 135”

“eRNA are unstable transcripts emerging from active enhancer, and acting as transcriptional regulators”

  1. P4, line 152: does KLF5 depletion also affect CFTR expression or only channel activity? Pleases precise

Response: We understand the reviewer advice and modified the text to well explain the impact of this depletion.

"Page 5, Lines 164-165

  1. P5, line 176: state that Caco2 cells are of intestinal origin and frequently used as a model

Response: We appreciate the Reviewer advice and agree that we could specify the use of these cells as intestinal model.

"Page 4, Lines 188

“Assuming that cooperation is frequently used, other combinations have been made in Caco2 cells, the most frequently intestinal cell line used.”

Reviewer 2 Report

The review is generally well written and is a useful summary for its topic, enabling readers to understand the molecular basis of tissue specific regulation of CFTR expression in several different tissues. As this is a review article and not a research article, I have few specific comments on the content of the manuscript, which the authors have covered quite comprehensively and logically, with appropriate references (apart from the last few sections where the focus seems to be lost: see last comments).

However, there are several mistakes and typographical errors and the manuscript should be read through carefully by the authors for corrections to be made. I give some examples here:

lines 7-8: “largely studied and care for individuals improved past years” should be rewritten to make sense.

line 11: “take part of” should be “take part in”

line 43: “canal” should be “channel”

line 54: “disorders” should be “disorder”

line 66: “in a lesser” should be “to a lesser”

line 108-9: “enhancer blocking” should be given the full name, “enhancer blocking insulator” – this omission also occurs later on in the paper: please check

line 108: “his role” = “its role”

line 121: “Histone marks study correlate”: the meaning of this is unclear, please clarify the phrase

line 143: “effect in” = “effect on”

line 207: “have” = “has”

Etc… There are several more of these kinds of changes that need to be made. I believe the authors will be able to detect and correct these before resubmission.

General comments on content.

Section 4.5: Modifier genes. This section could contain more detail about the tissue specific aspects of the effect of modifier genes. At the moment the section is little more than a list of different modifier genes associated with different tissues. However, it would be improved by some discussion of potential mechanisms for these associations, and perhaps their implications for CF disease.

In the following section (4.6), the relationship between complex alleles and tissue specificity should also be made clearer, for the section to be relevant.

Finally, it is also puzzling that the conclusions of the paper are not more focused on tissue specificity. An attempt should be made to summarize the paper and bring out some unifying conclusions which focus primarily on the topic stated in the title.

In general the english language is fine, but there are some mistakes, some omissions and some inconsistencies. I have pointed out a selection of these in my report, but please read the whole manuscript carefully and try to clean them all up.

Author Response

Comments and Suggestions for Authors

The review is generally well written and is a useful summary for its topic, enabling readers to understand the molecular basis of tissue specific regulation of CFTR expression in several different tissues. As this is a review article and not a research article, I have few specific comments on the content of the manuscript, which the authors have covered quite comprehensively and logically, with appropriate references (apart from the last few sections where the focus seems to be lost: see last comments).

However, there are several mistakes and typographical errors and the manuscript should be read through carefully by the authors for corrections to be made. I give some examples here:

lines 7-8: “largely studied and care for individuals improved past years” should be rewritten to make sense.

line 11: “take part of” should be “take part in”

line 43: “canal” should be “channel”

line 54: “disorders” should be “disorder”

line 66: “in a lesser” should be “to a lesser”

line 108-9: “enhancer blocking” should be given the full name, “enhancer blocking insulator” – this omission also occurs later on in the paper: please check

line 108: “his role” = “its role”

line 121: “Histone marks study correlate”: the meaning of this is unclear, please clarify the phrase

line 143: “effect in” = “effect on”

line 207: “have” = “has”

Etc… There are several more of these kinds of changes that need to be made. I believe the authors will be able to detect and correct these before resubmission.

Response: First of all, thanks to reviewer for the comments and suggestions that we are sure will improve our manuscript.

General comments on content.

  1. Section 4.5: Modifier genes. This section could contain more detail about the tissue specific aspects of the effect of modifier genes. At the moment the section is little more than a list of different modifier genes associated with different tissues. However, it would be improved by some discussion of potential mechanisms for these associations, and perhaps their implications for CF disease.

Response: We appreciate the Reviewer advice and agree that we could specify more the potential impact of modifier genes. We add some hypothesis but it is hard to speculate because very little functional tests have been performed in the different studies.

  1. In the following section (4.6), the relationship between complex alleles and tissue specificity should also be made clearer, for the section to be relevant.

Response: We thank the Reviewer for this comment and agree that the relationship between complex alleles and tissue specificity is not enough developed. We added these sentences in the revised manuscript.

"Page 10, Lines 389-393

  1. Finally, it is also puzzling that the conclusions of the paper are not more focused on tissue specificity. An attempt should be made to summarize the paper and bring out some unifying conclusions which focus primarily on the topic stated in the title.

Response: Reviewer comment is correct and very relevant, we tried to more focus our conclusions on tissue specificity.

  1. Comments on the Quality of English Language

In general the english language is fine, but there are some mistakes, some omissions and some inconsistencies. I have pointed out a selection of these in my report, but please read the whole manuscript carefully and try to clean them all up.

Response: We understand well this remark, and we take it account. We have carefully reviewed the manuscript and corrected the English grammar and syntax.
